# Porphyrin Macrocycles: General Properties and Theranostic Potential

**DOI:** 10.3390/molecules28031149

**Published:** 2023-01-23

**Authors:** Rica Boscencu, Natalia Radulea, Gina Manda, Isabel Ferreira Machado, Radu Petre Socoteanu, Dumitru Lupuliasa, Andreea Mihaela Burloiu, Dragos Paul Mihai, Luis Filipe Vieira Ferreira

**Affiliations:** 1Faculty of Pharmacy, “Carol Davila” University of Medicine and Pharmacy, 6 Traian Vuia, 020956 Bucharest, Romania; 2“Victor Babeş” National Institute of Pathology, 050096 Bucharest, Romania; 3Polytechnic Institute of Portalegre, 7300-110 Portalegre, Portugal; 4BSIRG—Biospectroscopy and Interfaces Research Group, iBB-Institute for Bioengineering and Biosciences, Instituto Superior Técnico and Associate Laboratory i4HB—Institute for Health and Bioeconomy at Instituto Superior Técnico, Universidade de Lisboa, 1049-001 Lisboa, Portugal; 5“Ilie Murgulescu” Institute of Physical Chemistry, Romanian Academy, 060021 Bucharest, Romania

**Keywords:** porphyrin macrocycles, photodynamic therapy, theranostic agents

## Abstract

Despite specialists’ efforts to find the best solutions for cancer diagnosis and therapy, this pathology remains the biggest health threat in the world. Global statistics concerning deaths associated with cancer are alarming; therefore, it is necessary to intensify interdisciplinary research in order to identify efficient strategies for cancer diagnosis and therapy, by using new molecules with optimal therapeutic potential and minimal adverse effects. This review focuses on studies of porphyrin macrocycles with regard to their structural and spectral profiles relevant to their applicability in efficient cancer diagnosis and therapy. Furthermore, we present a critical overview of the main commercial formulations, followed by short descriptions of some strategies approached in the development of third-generation photosensitizers.

## 1. Introduction

Interdisciplinary research on obtaining new pharmaceutical forms to guarantee, in one step, the identification of and therapy for certain tumor formations occupies a special place in the medical field and generates some top themes in terms of objectives targeting improvements in the health of the population. Despite specialists’ efforts to find the best solutions in this respect, cancer generally remains the biggest health threat on the planet. Characterized by an uncontrollable growth of cells that are able to invade any part of the organism, cancer is a conglomeration of several neoplastic diseases that can be triggered by a multitude of factors, both endogenous and exogenous. Nowadays, global statistics concerning deaths associated with cancer are alarming [1,2,3]. At the global level, about 19.3 million new cases were reported in 2020, and it was one of the main causes of death in most countries [4,5,6]. These statistics justify the need to intensify interdisciplinary research in order to identify efficient strategies for cancer diagnosis and therapy by the use of new molecules with optimal therapeutic potential and minimal adverse effects.

The integration of therapeutic and diagnostic potential into single drug molecules is one of the most important concerns of specialists in the biomedical field [4,7].

Porphyrins possess selectivity for tumor cells, low cytotoxicity in the absence of light, and peculiar photophysical properties (absorption and emission in the visible region, high triplet state quantum yield, high quantum yield of ^1^O_2_* generation), which have drawn research interest for their use as theranostic agents [8,9,10,11,12,13,14].

Compared with conventional antitumor therapies, which are known for their highly toxic effects on normal cells, photodynamic therapy (PDT) is noninvasive and has minimal side effects [15].

PDT is a selective method of antitumor treatment based on cell necrosis/apoptosis induced by a reactive species derived from molecular oxygen, generated during the irradiation of photosensitizer (PS) molecules in the presence of oxygen molecules. Tetrapyrrolic molecules such as porphyrins, chlorins, phthalocyanines, and bacteriochlorins are macrocyclic structures frequently used as photosensitizers. This review is focused on porphyrin macrocycles, and summarizes aspects of their structural and spectral profiles relevant to their applicability in cancer theranostics.

## 2. General Properties of Porphyrin Macrocycle

Porphyrins are among the most examined tetrapyrrolic macrocycle structures with applicability in the biomedical field. Starting at the beginning of the 20th century, it was revealed that the tetrapyrrolic macrocycles present an inherent photosensitive affinity characteristic in relation to tumor cells, proving their potential use in antitumor therapy [16,17,18,19]. The structural characteristics, spectral profiles, and great coordinating capacity of tetrapyrrolic molecules are the main factors behind their therapeutic potential.

Structurally, porphyrin molecules include four pyrrolic units bound by methine bridges (the C atoms in the 5, 10, 15, and 20 positions of the macrocycle, also named *meso*-positions) (Figure 1) [20]. The aromatic character of tetrapyrrolic macrocycles is an essential property of these structures, which is important in defining the spectral profile. The π electron system contains 22 electrons, among which 18 are delocalized and follow Hückel’s rule concerning the aromatic character. The presence of nitrogen atoms inside the macrocycle provides ligand properties for these molecules, allowing the possibility of coordinating metal ions [21,22].

Furthermore, the properties of metal ions (ionic volume and electronic configuration) may facilitate reactions of metalloporphyrin with monodentate ligands from the biologic environment [23]. Hemoglobin, cytochromes, catalase, and peroxidases are examples of molecules that contain metalloporphyrin structures and have an important role in biological processes [24,25,26,27].

The porphyrinic structures are versatile and can be shaped by attaching peripheral substituents with a polarity degree adequate for internalization in cells.

Their unique structural and spectral properties, especially their excellent photochemical and photophysical properties, define the profile of porphyrins as therapeutic candidates.

Porphyrins have a typical absorption spectrum described by an intense band (Soret band) located between 400 and 440 nm, and four Q bands (Q_y_(1,0), Q_y_ (0,0), Q_x_ (1,0), Q_x_ (0,0)) with reduced intensity and placed in the 440–800 nm spectral range (Figure 2). The Soret band is not relevant for photodynamic therapy of deeper tumor tissue; only the 600–800 nm range is useful for antitumor therapy by photosensitization.

In the case of metalloporphyrins, the absorption spectrum is described by the Soret band, placed in the 400–440 nm range, and one or two Q bands located in the 550–600 nm range (Figure 2).

The spectral profile of porphyrins is determined by the molecular symmetry, the nature and position of peripheral macrocycle substituents, and the nature of the metal ions in the case of metalloporphyrins [28,29,30].

**Figure 2 molecules-28-01149-f002:**
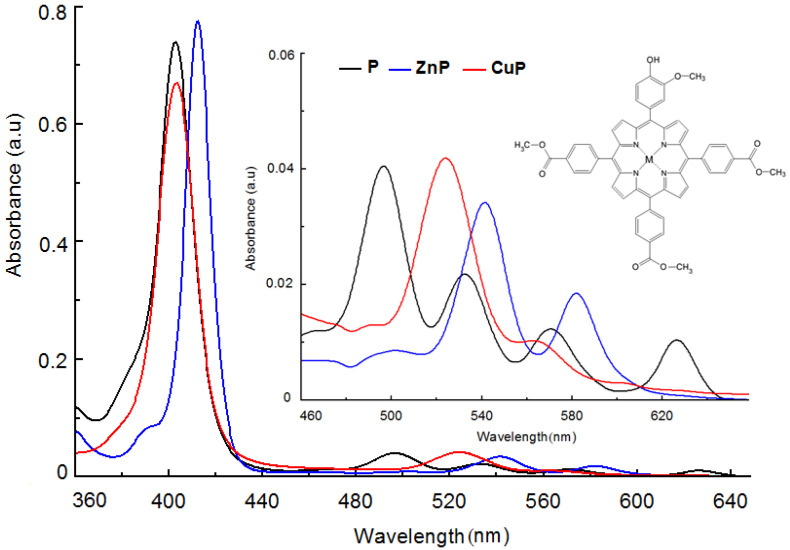
Absorption spectra for: 5-(4-hydroxy-3-methoxyphenyl)-10,15,20-tris(4-carboxymethylphenyl) porphyrin (P), 5-(4-hydroxy-3-methoxyphenyl)-10,15,20-tris(4-carboxymethylphenyl)porphyrinatozinc(II) (ZnP) and 5-(4-hydroxy-3-methoxyphenyl)-10,15,20-tris(4-carboxymethylphenyl)porphyrinatocopper(II) (CuP); (dimethylsulfoxide used as solvent) [31].

A ratio of the relative intensities of the Q bands IV > III > II > I describe, for the porphyrinic compound, a spectrum of absorption of *etio* type and the porphyrin fits the group of etioporphyrins [30]. This type of electronic spectrum is characteristic to the porphyrins, for which six or more of the positions β of the macrocycle have substituents with no π electrons.

Substituents with π electrons attached to the *β* position of the macrocycle determine spectral modifications by shifting the maximum absorption to a higher wavelength and modifying the ratio of Q band intensities, in the order III > IV> II > I.

Longer shifts to the red side of the absorption maximum are registered when substituents with double or triple bonds occupy the *β* positions and the ratio of Q band intensity is III > II > IV > I. If the *meso* positions are occupied, the Q band intensity decreases in the order IV > II > III > I [30].

Significant spectral modifications appear by protonation of the nitrogen atoms from the tetrapyrrole core. The acidic environment determines the protonation of the nitrogen atoms, which are inside the tetrapyrrole core, with the transition to the dicationic shape H_4_P^2+^ accompanied by the modification of the absorption spectrum, respectively, the reduction of the number of Q bands and red shifts of the absorption peaks [28,29]. Therefore, studies on the spectral influence of pH modifications in biological areas where these tetrapyrroles show activity are needed in terms of their potential therapeutic use.

Another factor that has a major influence on the spectral behavior of porphyrins is molecular aggregation, a phenomenon that depends on the structural characteristics, pH value, polarity, and ionic strength of the solution in which these compounds are activated [32,33]. The 22 π electrons of the porphyrin macrocycle facilitate strong π–π interactions and the formation of J-type or H-type aggregate, each leading to specific spectral features in terms of bathochromic or hypsochromic shifts in the corresponding absorption spectra.

The remarkable absorption and emission properties of tetrapyrrolic chromophores are the most relevant from the point of view of their biomedical applicability [34,35,36].

Porphyrins have a good fluorescence profile, with emission capacity in the 600–800 nm spectral range, and the potential for reactive oxygen species (ROS) formation by irradiation in the presence of molecular oxygen. The fluorescence spectrum of a porphyrinic compound (Figure 3) includes two spectral bands, as a result of the radiation emission that takes place during the transition of the molecule from the excited singlet state, S_1_, to the ground state, S_0_; these bands are located at longer wavelengths compared to the excitation wavelength (λ_emis_ > λ_abs_), due to the loss of energy through vibrational relaxation in the excitation state. The registered difference between λ_max_ of the absorption band and λ_max_ of the emission band is named the Stokes shift.

The specific parameters for molecular fluorescence evaluation are fluorescence lifetime (τ_s_) and fluorescence emission quantum yield (Φ_F_). They can be experimentally determined by spectral fluorescence analysis with the use of modern laser equipment; their test excitation time is in the picoseconds, and they have some rapid response photodiodes [37,38,39].

The fluorescence of the tetrapyrrole structures appears as a result of the aromatic character imprinted on the molecule by the π electrons of the double conjugated bonds. The delocalized electrons can be involved in π→π* transitions and go through singlet excitation states.

The fluorescence of porphyrins is influenced by the nature of the peripheral substituents of the macrocycle; some substituents, such as –NH_2_, –OH, and –O–CH_3_, have the effect of delocalizing the π electrons, with an increased probability of transitions between the excited singlet state and the ground state, and they have an effect on increasing the quantum yields. On the other hand, electron withdrawing groups (e.g., –NO_2_ or CO_2_H) lower or eliminate the fluorescence [40]. The values of the parameters that define the fluorescent profile of a tetrapyrrole (lifetime of fluorescence (*τ*_s_) and fluorescence efficiency (Φ_F_)) are modified by complexation with metal ions, and the modifications are dependent on the electronic structure of the cation. With the complexation of metal ions with high volume (e.g., Pd^2+^), the values of Φ_F_ and τ_s_ decrease. The efficiency of singlet oxygen species formation also decreases [39].

Porphyrinic compounds with paramagnetic metal ions (Fe^2+^, Cu^2+^, Co^2+^) present phosphorescent properties, while the lifetime of the excited singlet state is very short [41,42].

## 3. Aspects regarding Porphyrin Macrocycles as Photosensitizers

### 3.1. Short Background regarding to the Porphyrinoid Photosensitizers

Among the first clinical observations on the photosensitization effect of a chemical compound (eosin) on the tumor cells are those made from 1903 by the pharmacologist Hermann von Tappeiner and the dermatologist Albert Jesionek. By topical application of the eosin (photosensitive dye) and exposure of the treated area to light, the two researchers realized for the first time the photodynamic therapy and they introduced the term of “photodynamic reaction” [43].

Subsequent studies have shown that hematoporphyrin (Hp), combined with ultraviolet light, has a therapeutic potential on the skin diseases, including psoriasis [44,45,46,47].

The purification of hematoporphyrin led to the HpD, a combination of monomers and oligomers with tetrapyrrole structures, with remarkable fluorescent properties and affinity for the tumor cells. HpD was used for the first time by R. L. Lipson as an imagistic imaging agent to visualize the tumor lesion during surgery [45,48]. Later, the team led by Thomas J. Dougherty purifiedHpD and obtained a heterogenous mixture of porphyrins, which was marketed as Photofrin [49,50].

The clinical use of Photofrin was limited by its difficult accumulation at cellular level, weak absorption at λ = 630 nm, and the patients’ photosensitivity to the sun for a long period of time after the treatment [51,52].

The deficiencies appearing in the clinical usage of HpD and Photofrin led to the development of new porphyrinoid photosensitizers with unique structure, high absorption in the visible-near infrared spectral domain and high singlet oxygen quantum yield. They were classified as second generation photosensitizers, and include the porphyrin and porphyrin analogue photosensitizers (chlorin, (iso)bacteriochlorin, phthalocyanines) such as Hemoporfin^®^, Foscan^®^, Visudyne^®^, Tookad^®^, Radachlorin^®^, Photochlor^®^, and Photocyanine^®^ [53,54].

However, after years of development, the second-generation porphyrin photosensitizers had some disadvantages which include reduced solubility in water and biological fluids, difficult location at the cell level, the molecular aggregation tendency, and the effect of reducing the photodynamic efficiency [55,56,57].

In recent years, researchers have obtained and clinically investigated a new generation of photosensitizers, the so-called third-generation photosensitizers, characterized by good solubility in biologic fluids, great capacity for generating reactive oxygen species (ROS) and good selectivity in relation to tumor cells [58,59,60].

The synthesis of third generation PS is carried out in a chemo- and regioselective manner, by the attachment of fragments of bioactive molecules (folic acids, peptides, sterols, proteins, anti-tumor monoclonal antibodies, sugars), polyethyleneglycol or nanoparticles to the structures from the second-generation porphyrin PS [55,56,57,61,62,63,64,65,66]. These porphyrin nanophotosensitizers are currently in the development stage and have not been used in clinical trials.

Due to their structural shape and unique affinity for tumor cells, porphyrins can be used as excellent carriers to transport anticancer drugs into tumor tissues, with synergistic effects in the identification and destruction of tumors. Therefore, current research based on the development of porphyrins as theranostic agents mainly only includes studies on porphyrins as anticancer drugs and porphyrins-bonded anticancer drugs.

### 3.2. Basic Photochemical Principles in PDT

Establishing a framework protocol to completely cure cancer by using optimal pharmacological molecules with minimal adverse effects is one of the prime objectives of research studies in the pharmaceutical and oncological fields. Tetrapyrrolic macrocycles are among the most studied families of compounds with regard to their therapeutic potential, especially their antitumor potential. Through their structural profile and electronic load, these structures can efficiently generate reactive oxygen species (ROS), such as singlet oxygen, in the presence of molecular oxygen and light.

Clinical experiments of irradiating porphyrinic structures (at the cell level) in the presence of molecular oxygen laid the foundation for a modern noninvasive treatment method: photodynamic therapy (PDT) [16,17,67].

The clinical procedure is initiated by administering a photosensitizer (PS), followed by light irradiation at a certain wavelength, usually a red light laser, after the accumulation of PS in the tumor tissue. Light irradiation has an important role in photodynamic therapy, because it offers the necessary energy to produce reactive species, which destroy tumor cells. Moreover, molecular oxygen has a determinant role in initiating and propagating the reactions of destroying tumor formations [68].

The activation of PS molecules by irradiation generates a series of competitive reactions of ROS, which induce necrosis and/or apoptosis of tumor cells. Apoptosis is a form of programmed cell death (the cells start their own destruction) as a response to the physiological key index and intracellular injury. During apoptosis, cell collapse is characterized by key morphological and chemical modifications, followed by phagocytosis. These modifications take place while maintaining the levels of adenosine triphosphate, activating caspase (an enzyme with proteolytic activity and a predominant role in the process of self-destruction), chromatin condensation, and the fragmentation of DNA and formation of apoptotic bodies. During apoptosis, an associated inflammatory response is not identified [69].

Necrosis represents a form of uncontrolled cell death in response to a series of factors, such as physical injury, ischemia, excessive ROS accumulation, or the presence of infectious agents. Unlike apoptosis, necrosis is a local inflammatory response seen as a result of cells being liberated directly in the surrounding tissue [69].

Schematically, the photophysical processes that form the basis of generating ROS are rendered in Figure 4 (Jablonsky diagram).

In the initial stage, by absorbing light radiation, the molecules of the photosensitizer go from the ground state (S_0_) to a superior energetic state, the excited singlet state (S_1_); the transformation takes place rapidly (~10–15 s) through the transition of electrons on the S_0_ level to the S_1_ excited singlet state. The duration of the S_1_ state is in nanoseconds. Stabilization of the S_1_ state can be reached by the emission of photons (fluorescence) or by a process called intersystem crossing (ISC), by which the electronic spin is reversed and the electron passes onto an energetic level (T_1_) of the excited triplet state [70,71,72]. The energy of T_1_ species can be dissipated by the passage to a fundamental state with weak light emission [70,71,72]. The lifetime of the T_1_ species is in micro- to milliseconds, much longer than the lifetime of the excited singlet species; consequently, it is more likely that the PS found in the excited triplet state participates in photodynamic reactions. Thus, the photodynamic effect is the result of the energetic or electronic transfer of the photosensitizer found in the T_1_ state on the way to the organic substratum or molecular oxygen. The stabilization reaction of the excited triplet state (T_1_) of the photosensitizer can occur by two types of mechanisms, classified as reactions of type I and II [69].

Type I reactions (free radical mechanism) are reactions of oxidation and reduction and imply the transfer of electrons and hydrogen atoms between the photosensitizer (found in the excited singlet or triplet state) and the biomolecules, resulting in free radical formation; these radicals interact with oxygen molecules, resulting in the formation of superoxide anion radicals, which can generate forward hydroxyl radicals and may determine the formation of cytotoxic species.

Type II reactions (singlet oxygen mechanism) take place through the interaction of photosensitizer molecules in the excited triplet state and molecular oxygen (triplet state) with the formation of singlet oxygen, a very reactive species in relation to the cell components (lipids, proteins, and nucleic acids) with a large potential to destroy tumor cells [73,74,75,76,77,78,79]. Moreover, type II reactions proceed without the chemical transformation of PS molecules, which is why the set of photochemical reactions can be remade with a new activation. As the half-life of the reactive species is very short, fewer than 40 ns, the photodegradation reactions unfold next to the areas of initiation of the photochemical reactions, which is why the PS location at the cell level is determined through its photodynamic reactions [15,80,81].

### 3.3. Chemical and Pharmaceutical Aspects Concerning Photosensitizers with Tetrapyrrolic Structures Used in Diagnosis and Antitumor Therapy

The theranostic agent profile assigned to an active substance defines its potential to act simultaneously as a marker for the detection of tumor cells and as a therapeutic agent [8].

Photodynamic therapy (PDT) uses porphyrins as photosensitizers, which, by their structural and spectral properties, have superior pharmacological potential compared to classic antitumor molecules [9,82,83]. At present, PDT is an efficient therapeutic method successfully applied for the treatment of tumor formation in the skin, breast, lung, throat and brain [84,85,86,87]. Photodynamic therapy is a selective method of antitumor treatment based on the cell necrosis/apoptosis induced by ROS generated during the irradiation of photosensitizer molecules in the presence of oxygen molecules. The phototoxic reactions induced by PDT are limited to the tumor area, where the photosensitizer accumulates and the irradiation takes place, and for this reason, PDT has limited side effects compared to the standard treatment methods [88]. Lately, the scientific community has made significant investments into developing and implementing new active molecules in oncology that are both therapeutic and diagnostic [89,90,91,92]. Regardless of the therapeutic purpose, ensuring optimal biomedical efficiency requires that the photosensitizer meets the following mandatory requirements [72,82]:It has a unique, well-defined structure with maximum purity and can be obtained by modern ecological methods;It has a structural profile that allows optimal internalization at the tumor level and is described by a well-defined distribution of lipophillic/hydrophilic groups at the periphery of tetrapyrrolic macrocycles;It has a well-defined spectral profile with maximum absorption in the therapeutic range (600–850 nm) associated with good efficiency in generating singlet oxygen (Φ_Δ_ ≥ 0.5) and a triplet state with a lifetime in the microsecond range;It is soluble in nontoxic solvents accepted for pharmaceutical formulations;It is nontoxic in the absence of light;It provides rapid clearance from the organism;There is no toxic effect with the necessary dose for a therapeutic effect;There is an absence of toxicity for the photosensitizer metabolites.

The molecular structure of the photosensitizer is the predominant factor in its pharmacokinetic and pharmacodynamic evolution, no matter the type of therapeutic approach. Biodistribution at the tumor mass level has direct consequences in terms of the pharmacodynamic efficiency of any active substance [53]. Among the factors that influence the intracellular uptake of PS are its solubility in biological fluids and the amphiphilic character of the molecule, which determines the penetration of the double-lipid stratum of the cell membrane.

By evaluating the relationship between the PS structure and the biodistribution capacity at the tumor tissue level, Boyle and Dolphin classified PS into three main groups [53]:Hydrophobic photosensitizers, which have peripheral substituent hydrophobic functional groups and very reduced solubility in polar solvents, alcohol, or water, at a physiological pH;Hydrophilic photosensitizers, which have three or more peripheral functional substituent hydrophilic groups and slight water solubility at physiological pH;Amphiphilic photosensitizers, which have in their structure hydrophobic and hydrophilic functional groups and water solubility at physiological pH.

The solubility of tetrapyrrolic compounds meant for biomedical applications represent a current problem in the field of pharmaceutical chemistry, because the clinical efficiency of the active substance is directly influenced by its solubility in the biological fluid and membrane environment [93]. For example, chlorin-type macrocycles, although characterized by stronger absorption capacity in the red area of the visible spectrum, present the disadvantage of limited water and alcohol solubility. Another disadvantage of tetrapyrrole-type photosensitizers is the molecular aggregation tendency, with the effect of reducing the photodynamic efficiency, specifically the efficiency of producing reactive singlet oxygen and the fluorescence lifetime [58,76].

At present, the efforts of specialists who work in medicinal chemistry are oriented toward synthesizing photosensitizers with an amphiphilic structure by introducing peripheral substituents (lipophillic and hydrophilic) at the periphery of the macrocycle.

Some modern synthesis methods introduce, as substitutes, residues of polyethylene glycol in order to obtain a soluble structure in water with excellent photophysical properties [94,95,96,97,98].

A series of photosensitizers with macrocycle structures approved or in clinical trials for diagnosis and photodynamic therapy of cancer, are presented in Table 1.

#### 3.3.1. PHOTOFRIN^®^

Photofrin^®^ (Axcan Pharma, Inc.) is a first-generation photosensitizer used in photodynamic therapy. Chemically, it is a mixture of monomers, dimers, and oligomers of hematoporphyrin, as well as esters and ethers corresponding to these compounds. The preparation of this mixture from derivatives of hematoporphyrin was reported for the first time by Dougherty in 1983 [49].

Spectrally, phosphate-buffered saline solution containing Photofrin^®^ presents in the spectrum of molecular absorption at a maximum 630 nm (ε = 3.0 × 10^3^ M^−1^ cm^−1^) and has lower efficiency of singlet oxygen generation (Φ_Δ_ = 0.01). In clinical applications, Photofrin^®^ is given as an intravenous injection at a dose of 2–5 mg/kg, and after 24–48 h it is activated by light irradiation at *λ* = 630 nm (light dose 100–200 J cm^−2^).

Photofrin^®^ was approved by the Canadian Health Agency in 1993 for antitumor therapy against bladder cancer [49]. The therapeutic applicability of Photofrin^®^ was later extended for the treatment of lung, esophageal, bladder, and early stages of cervical cancer [73,74,76,105]. Photofrin^®^ is used to identify and treat tumor formations at the brain and lung level, and cervical dysplasia [9,102]. Wilson et al. used Photofrin^®^ as a marker to identify tumor formations at the brain level and as an antitumor agent for the treatment of waste tumor cells after surgery [106,107,108,109]. The results obtained by using PDT with Photofrin^®^ were explained by a series of disadvantages, mainly determined by the chemical composition of the PS (mixture of chemical compounds, some of them inactive PDT). The non-homogeneous composition of PS determines a difficult location at the tumor mass level, with weak radiation absorption at long radiation time and activation.

The most important disadvantage of Photofrin^®^ is the long photosensitization after treatment, due to which patients must be protected against bright light for 6–12 weeks. The ability to repeat the treatment is limited by these disadvantages [17,51,52].

Other pharmaceutical forms that contain, as active substances, mixtures of monomers, dimers, and oligomer structures derived from hematoporphyrin are Photogem^®^ and Photosan^®^. Photogem^®^ was approved for clinical use in Russia and Brazil, and Photogem^®^ was clinically approved in the European Union [110,111,112].

#### 3.3.2. FOSCAN^®^

The tetrapyrrole compound 5,10,15,20-tetrakis(3-hydroxyphenyl)chlorin, known as temoporfin or Foscan^®^ (the commercial name), is one of the strongest photosensitizers used in photodynamic therapy against cancer. Temoporfin has a symmetric structure defined by identical substitutes of the four *meso* positions of the tetrapyrrole macrocycle, and it was obtained and characterized in terms of its toxicological and physico-chemical aspects by Bonnet et al. [113]. It is a dark violet crystalline powder, insoluble in water but soluble in alcohol, acetone, and ethyl acetate.

The structure of 5,10,15,20-tetrakis(3-hydroxyphenyl)chlorin is characterized by a maximum absorption capacity at 652 nm (*ε*~35,000 M^−1^ cm^−1^) and good distribution at the tumor tissue level. The photophysical properties of temoporfin include a quantum yield of singlet oxygen generation Φ_Δ_= 0.4 (measured in dimethylsulfoxide) and high efficiency in initiating phototoxic reactions.

Foscan^®^ (Biolitec Pharma Ltd., Jena, Germany) is a solution for injection that contains, as an active substance, temoporfin 4 mg/mL with ethanol solvent and propylene glycol co-solvent. The commercial formula does not contain water, because the stability of temoporfin decreases in an aqueous environment.

In clinical applications, Foscan^®^ is intravenously administered at a dose of 0.15 mg/kg and then activated by laser light irradiation at a wavelength of 652 nm about 98 h later. An energy dose of 10–20 J cm^−2^ is used for irradiation. Foscan^®^ is administered only at oncology centers under the medical supervision of a photodynamic therapy specialist. The therapeutic effect is mediated by generating reactive oxygen species, resulting in intracellular interactions between temoporfin, light, and molecular oxygen. By restrictive exposure of tumor formations to laser radiation, the cell destruction by reactive oxygen species is limited to the components of the tumor cells. Foscan is an antitumor agent that was approved in 2001 for photodynamic therapy of advanced squamous cell carcinoma at the brain and throat level (a form of cancer that starts at the nasal, throat, and ear membrane level), and is administered to patients who do not respond to chemotherapy or radiotherapy.

Foscan^®^ has applicability in antitumor diagnosis, as a marker in identifying the modifications that appear in bladder cancer, and in the diagnosis of ovarian cancer and tumors at the brain level [100,114,115,116]. It was clinically investigated for use in breast, pancreas, lung, stomach, and skin cancer [63,99].

Temoporfin is a second-generation photosensitizer and, under the same experimental conditions (photosensitizer dose, energy dose), presents superior clinical efficiency (by approximately 100 times) to the first-generation Photofrin^®^ [117].

A disadvantage of temoporfin is the long removal time from plasma (up to 4–6 weeks after intravenous administration), which induces photosensitivity in patients, similar to Photofrin^®^ [118]. Further disadvantages include side effects such as constipation, difficulty digesting food, necrotizing stomatitis, and facial edema.

To obtain a complete response to treatment, homogeneous distribution of the PS molecule at the tumor formation level is required, which is dependent on the pharmaceutical formula in which the agent is administered.

The problems of solubility and transport to the cells are topics under investigation in pharmaceutical research, constituting a major objective for introducing agents to pharmacokinetically improve the controlled and selective delivery of PS to tumor cells while reducing toxicity to healthy tissue [119].

That is why, in order to optimize the response to antitumor therapy, other medicines that contain temoporfin were included: Foslip^®^ and Fospeg^®^ (Biolitec Pharma Ltd., Jena, Germany), which have demonstrated a better ability to internalize the PS in the tumor mass, with better clinical effects compared to Foscan^®^ [120]. The experimental data prove that Foslip^®^ and Fospeg^®^ have efficient photosensitivity similar to Foscan^®^, but reduced cytotoxicity [121,122].

Foslip^®^ contains temoporfin conditioned in conventional liposomes. Fospeg^®^ has temoporfin expressed in liposomes covered with polyethylene glycol (PEG). It presents pharmaceutical properties superior to Foscan^®^ (demonstrated by studies carried out on rats with tumors), and allows the administration of a third Foscan dose in order to induce necrosis in the whole tumor volume [123].

Recent in vivo comparative studies by Reshtov et al. reported better clinical efficiency for the liposomal formula with PEG, with reduced time between administration and irradiation of PS at the skin level, which significantly reduces the side effects at the skin level [124]. Moreover, biological evaluation of the two liposome formulae (Foslip^®^ and Fospeg^®^) has followed the internalizing potential, with the distribution of PS at the tumor mass level correlated with photosensitizing efficiency [119,123,124,125].

#### 3.3.3. LASERPHYRIN^®^

Mono-l-aspartyl chlorin e-6 is a second-generation photosensitizer obtained by the interaction of dicyclohexylcarbodiimide of chlorin e-6acid and di-*tert*-butylaspartate [126]. It is a dark green, water-soluble crystalline powder. The absorption spectrum of mono-l-aspartyl chlorin e-6 is typical of chlorine type compounds, and it has generated interest for its clinical applications (with a maximum absorption band at 654 nm; ε = 4.0 × 10^4^ M^−1^ cm^−1^). The molecule’s good efficiency in generating singlet oxygen is shown by the parameter of quantum efficiency, Φ_Δ_ = 0.77 (Φ_Δ_ determined by photoirradiation in a buffer solution of phosphate) [63,113].

Talaporfin has a shorter accumulation time at the tumor level, more rapid clearance, and better biodisponibility than Photofrin^®^.

Laserphyrin^®^ (ME2906; Meiji Seika Pharma Co., Ltd., Tokyo, Japan) was approved in 2004 for clinical use in photodynamic therapy for lung cancer. At present, it is under clinical investigation for the treatment of colorectal neoplasm and brain, throat, and liver tumors. In clinical applications, PDT with Laserphyrin^®^ follows a procedure similar to the one used for Photofrin^®^, but the intravenous administration uses a smaller photosensitizer dose (0.5–3.5 mg/kg) and irradiation is performed at 664 nm 4 h after that dose. The primary action mechanism of mono-l-aspartyl chlorin e-6, by the effect of photosensitization, implies the degradation of tumor formation by vascular stasis and the direct effects of tumor cytotoxicity [127].

Based on the fluorescence properties and the observed internalization ability at the tumor level, the latest studies have demonstrated the clinical usefulness of Laserphyrin^®^ as an intraoperative photodiagnostic agent for brain tumors. The histological degree of malignity was correlated with the intensity of fluorescence and the concentration of photosensitizer in the tumor tissue.

Intravenous administration of 40 mg/m^2^ Laserphyrin^®^ 24 h before surgery and irradiation with laser light at 664 nm along with the surgery have allowed the guided resection of malignant tumors, and the results obtained have shown the theranostic agent profile of Talaporfin [104].

#### 3.3.4. VISUDYNE^®^

Chemically, verteporfin is a benzoporphyrin derivative, which was obtained in 1982 by Dolphin et al. by the reaction of the protoporphyrin dimethylester with dimethyl acetylenedicarboxylate [128]. The two regiomer forms (±) have similar pharmacokinetic properties concerning accumulation and clearance, which is why verteporfin has been considered from the pharmacologic point of view.

Verteporfin is a dark green, water-insoluble powder; in clinical applications, it is used in some formulae that contain lactose, dimyristoylphosphatidylcholine, sodium phosphatidylglycerol, ascorbyl palmitate, and butylhydroxytoluene. AsVisudyne^®^ precipitates in sodium chloride or other parenteral solutions, when preparing perfusion solutions saline is not used; instead, it is perfused with 5% dextrose solution and standard lines of perfusion, with hydrophilic membrane filters with pore size of at least 1.2 μm.

Visudyne^®^ is a second-generation photosensitizer. It was developed by QLT Phototherapeutics (Vancouver, BC, Canada) and was clinically tested in collaboration with Ciba Vision Corporation (Duluth, GA, USA) for the photodynamic treatment of exudative macular degeneration (wet) with subfoveal choroidal neovascularization, cutaneous membranes, and psoriasis [63]. Since 2001, it has been used as therapy in over 70 countries. Presently, Visudyne^®^ is manufactured by Novartis Pharmaceuticals (East Hanover, NJ, USA) in the form of a sterile powder for intravenous perfusion.

From the spectral point of view, Verteporfin has maximum absorption at 686 nm, with a molar absorption coefficient of *ε*~3.4 × 10^4^ M^−1^ cm^−1^. The singlet oxygen generating efficiency is Φ_Δ_ = 0.7 (methanol solvent).

The intravenous dose is 6 mg/kg, 30 min before irradiation, with an energy dose of 100 J/cm^2^. Activation of the photosensitizer takes place at *λ* = 686 nm. This photosensitizer rapidly accumulates in tumors (30–150 min after intravenous administration), while the remaining photosensitivity lasts only for several days, minimizing the photosensitivity effect on the patient [129].

Verteporfin registered accumulation and a clearance rate in the tumor mass 20 times greater than Photofrin^®^. The plasma half-life by the clearance of Verteporfin is approximately 6 h. Moreover, the red shift of the maximum absorption compared to Photofrin^®^ gives the advantage of better irradiation of tumor tissue. At approximately 690 nm, the tissue penetration power of light radiation is 50 times bigger compared to Photofrin^®^ [130,131,132]. Visudyne^®^ presents the advantage of spectral superiority to photosensitizers with second-generation porphyrinic structures (Foscan^®^, Photochlor^®^, and Laserphyrin^®^).

The latest studies have demonstrated the clinical efficiency of Visudyne^®^ in pancreatic tumor treatment, even in advanced stages of the disease [133]. Visudyne^®^ is a photosensitizer with potential as a theranostic agent, under investigation with promising results in ovarian cancer [134].

Among the adverse reactions to Visudyne^®^ administration, the most frequently reported are vision disorders, sensitivity in the region where the perfusion is administered (inflammation, pain, edema), and, rarely, allergic reactions.

#### 3.3.5. PHOTOCHLOR^®^

HPPH is a second-generation photosensitizer with a macrocycle structure, lipophilic character, and better internalization at the cell level. Pandey et al. reported on the preparation of HPPH from methyl pheophorbide, a derivative extracted from spirulina [135,136].

The spectral behavior of HPPH is relevant for biomedical applications. In micellar solution with 1% Tween-80, HPPH registers an absorption maximum at 665 nm, an extinction molar coefficient *ε*~4.75 × 10^4^ M^−1^ cm^−1^, and singlet oxygen formation quantum yield Φ_Δ_ = 0.48 (determined in dichloromethane). Through its molecular structure, Photochlor^®^ was proved to have some spectral characteristics and a certain amount of tumor accumulation superior to Photofrin^®^ or Foscan^®^ [117,137]. After intravenous administration, HPPH is selectively located in the tumor cell cytoplasm and presents a pharmacokinetic and pharmacotoxicological profile superior to Photofrin^®^ [135,136].

In clinical applications, Photochlor^®^ is administered intravenously at a dose of 0.15 mg/kg. Irradiation takes place at 665 nm, within 24–48 h from administration. Similar to Photofrin^®^, Photochlor^®^ is not metabolized, but is cleared from human plasma and slowly excreted [138].

Photochlor^®^ was investigated at the Roswell Park Cancer Institute (Buffalo, NY, USA) in stage I/II clinical studies of tumor formations at the esophagus level, in the mouth cavity, at the lung level, and for cervical intraepithelial neoplasm.

The use of Photochlor^®^ in PDT did not show any photosensitivity in patients after treatment, but it is recommended to avoid solar exposure for 7–10 days after administration [139]. The capacity to internalize at the tumor mass level and the fluorescence properties associated with the HPPH structure allow the use of this photosensitizer as a marker for the identification of different types of cancer [140].

#### 3.3.6. PURLYTIN^®^

Sn(II) ethyl-ethiopurpurin is a macrocyclic photosensitizer with Sn(II) metal ion. The compound was obtained and characterized by Morgan et al. and formulated and commercialized by Miravant Medica Technologies (Santa Barbara, CA, USA) [101,102,103,141].

The Sn(II) ethyl-ethiopurpurin complex occurs in the form of a water-insoluble powder, and for clinical application as a photosensitizer, a liposomal formulation is required.

Spectrally, the complex presents maximum absorption at 660 nm in the therapeutic field, thus assuring deep penetration in the tissues during irradiation. Its molecular absorption coefficient is ε~26,400 M^−1^ cm^−1^, and the singlet oxygen formation quantum yield, measured in acetonitrile, is Φ_Δ_ = 0.70 [142].

In clinical applications, Purlytin^®^ is administered intravenously as a liposomal solution at a dose of 1.2 mg/kg body weight. Activation by irradiation is carried out 24 h after administration, with laser light of 660 nm.

Purlytin^®^ is under investigation in stage I/II clinical studies for PDT applied to breast cancer, Kaposi’s sarcoma in AIDS patients, prostate cancer and brain metastases, and treatment of psoriasis. There have been no reported adverse side effects correlated with systemic toxicity with the administration of Purlytin^®^, but induced photosensitivity was observed in patients treated for a period of two weeks [101,102].

#### 3.3.7. TOOKAD^®^

Padoporfin has a complex combination structure of a macrocycle bacteriochlorophyll ligand with Pd(II) as the metal ion. It is manufactured by Negma Lerads/Steba Biotech, Toussous le Noble, France, and is used in stage II/III clinical studies of photodynamic therapy against prostate cancer [143,144,145,146].

The absorption spectrum of Padaporfin presents a band in the therapeutic range with a maximum at 762 nm and a molar extinction coefficient of *ε*~8.85 × 10^4^ M^−1^ cm^−1^. Consequently, it can be activated with light rays at 762 nm, i.e., deep tissue penetrating radiation. From the photophysical point of view, Padaporfin generates reactive species with good quantic efficiency of Φ_Δ_ = 0.50 (measured in dimethylsulfoxide) [101,102].

The clinical procedure for PDT mediated by Tookad^®^ is similar to that described for Photofrin^®^; it is administered intravenously at a dose of 2–4 mg/kg, and the irradiation takes place half an hour after administration.

The photosensitizer molecule has a strong lipophilic character, and the advantage of rapid clearance from the organism (less than 20 min) without inducing cutaneous phototoxicity compared to Visudyne^®^ and Lutrin^®^ [101,102].

#### 3.3.8. LUTRIN^®^

Texaphyrin lutetium or Motexafin lutetium (Lu-Tex) (commercial name Lutrin^®^) is a second-generation photosensitizer that was synthesized in 1994 by Sessler et al. by a condensation reaction of the derivatives of diformyltripyrrane and phenylenediamine [147]. The structure of the complex combination is characterized by water solubility and high selectivity in relation to tumor formation.

The spectral profile of the photosensitizer is described by absorption in the visible field at a maximum of 732 nm and a molar absorption coefficient of *ε* = 4.2 × 10^4^ M^−1^ cm^−1^ (determined in methanol solution). From the photophysical point of view, the capacity of Lutrin^®^ to generate reactive species of singlet oxygen is lower compared to other second-generation photosensitizers; in methanol solution, the singlet oxygen quantum yield is Φ_Δ_ = 0.11.

The clinical photodynamic approach for antitumor therapy follows a procedure similar to that for Photofrin^®^, with a compound dose of 0.6–7.2 mg/kg and activation at 732 nm, 3 h after administration.

Lutrin^®^ was approved for clinical use in therapy for recurrent prostate cancer and uterine and cervical cancer, and is in stage I/II/III clinical studies for breast cancer, melanoma, and Kaposi’s sarcoma [101,102]. The results obtained in the clinical evaluation of the therapeutic effect of Lutrin^®^ according to the compound dose, wavelength, and energy used in photochemical activation have confirmed good clinical efficacy by using a larger dose of the medicine (2 mg/kg) [148]. Clinical studies have also revealed some effects of photosensitivity induced by photodynamic therapy with Lutrin^®^ in patients with prostate cancer. The ligand of the macrocycle texaphyrin structure, together with Gd (III) ion, generates a complex combination (Xcytrin™) with applicability in tumor diagnosis and as an agent with radiotherapy potential and a chemosensitizer [149,150].

Xcytrin™ was investigated in stage III clinical studies for the treatment of primary tumor formation at the brain level. Xcytrin™ is characterized by an excellent potential for locating tumor cells compared to healthy ones, and through photoactivation it generates reactive species in the presence of intracellular oxygen and determines cell death by apoptosis [101,102,151].

#### 3.3.9. PHOTOSENS^®^

Photosens^®^ is a mixture of complex structures with photosensitizing ability. Chemically, the basic structure is represented by a complex combination of a central ion Al(III) and a macrocycle phthalocyanine ligand; the peripheral substitutes of the macrocycle are –SO_3_H groups. In the absorption spectrum, it presents a Q band with a maximum 676 nm and a molar absorption coefficient of *ε*~20 × 10^4^ M^−1^ cm^−1^.

Photosens^®^ has the largest molar absorption coefficient among all second-generation photosensitizers. The protocol for the clinical application of photodynamic therapy is similar to that applied for Photofrin^®^, with a reduced active substance dose. Photosens^®^ is administered intravenously at a dose of 0.5–0.8 mg/kg, and is activated by light at 150 J/cm^2^ at 676 nm 24–72h after administration.

Photosens^®^ is produced by the Niopik pharmaceutical company (Moscow, Russia) and was evaluated in stage III clinical studies for the treatment of skin cancer with squamous cells, breast tumors, and oropharyngeal and lung cancer. In order to improve the degree of internalization at the tumor level, Swiss clinicians have developed a liposomal formulation of phthalocyanine with Zn(II) (CGP55847) that they evaluated in stage III clinical studies for the therapy of carcinoma with squamous cells [101,152].

## 4. Strategies Aimed at Improving the Therapeutic Potential of Porphyrin Photosensitizers

With all of the advantages that second-generation photosensitizers have, including their possible double role as markers and therapeutic agents, it is a priority for specialists in oncology and medicinal chemistry to identify new structures with optimal pharmacological potential to assure the most advantageous cost/benefit ratio with regard to improving the health of the population.

Simultaneously with the use of second-generation photosensitizers in therapy, in recent years, through structural functional modifications of tetrapyrrolic macrocycles, researchers have obtained and clinically investigated structures characterized by good solubility in biologic fluids, great capacity for generating reactive oxygen species (ROS), good selectivity in relation to tumor cells, good clearance, and the absence of side effects associated with administration. These structures define a new generation of photosensitizers, the so-called third-generation photosensitizers, with pharmacological profiles superior to those of the second generation. The design and synthesis of third generation PS are carried out in a chemo- and regioselective manner, taking into account the set of mechanisms that govern the identification and destruction of tumor cells.

The main disadvantages of second-generation photosensitizers include reduced solubility in water and biological fluids, difficult location at the cell level, reduced clearance rate from the organism, molecular aggregation tendency, and the effect of reducing the photodynamic efficiency [102,103,104].

Presently, research based on development porphyrins in the oncological field, mainly includes two type studies: porphyrins as theranostic agents, and porphyrins-bonded anticancer drugs. Regarding the first type of study, in this section we will briefly describe some strategies aimed at improving the therapeutic potential of porphyrin as theranostic agents.

### 4.1. Design and Synthesis of New Unsymmetrical Porphyrins for Theranostics Applications

Numerous studies have reported that structural modifications by attaching functional groups with different polarities to the macrocycle, or introducing metal ions into the tetrapyrrole macrocycles, increase the stability and diminish the aggregation tendency of PS molecules. On the other hand, attaching functional groups with rich π electron content to peripheral macrocycles determines the red shift of maximum PS absorption together with spectral optimization [153,154,155,156,157,158,159].

Among the macrocycle structures, at present, porphyrins are the most studied in terms of architectural modifications with improvements in the spectral and pharmacologic profiles.

In recent years, our research group synthesized a panel of new amphiphilic porphyrins by ecological and versatile technological approaches in the framework of “*green chemistry*” (solvent-free reactions activated by microwave irradiation), as efficient strategies in the synthesis of new photosensitizers [31,160,161,162,163,164,165,166,167,168,169,170,171,172,173,174,175,176,177,178].

In the present study, we reviewed some recently reported unsymmetrical porphyrins as potential theranostics photosensitizers for cancer. These structures, by their advantages (good cellular uptake and appreciable biocompatibility in relation to tumor cells, appreciable fluorescence properties, and good singlet oxygen generation capacity), could provide a useful reference for the development of theranostics tools.

The particular profile determined by the presence of polar groups such as –OH, –OCOCH_3_ and –COCH_3_, enhances their ability to act as weak intermolecular physical bond generators, with increasing solubility of PS in polar environments.

By implementing drug design strategies, research has been aimed at obtaining unsymmetrical porphyrins with functional groups with acceptable volume that do not overly increase the molecular mass of the photosensitizers. Taking into account that these compounds must cross the cell membrane, the hydrophilic/lipophillic character must not be lost, and this was why we selected A3B and A2B2 porphyrins (Table 2) [31,160,161,162,163,164,165,166,167,168,169,170,171,172,173,174,175,176,177,178].

Among the most important advantages offered by these structural configurations is the excellent solubility of PS in PEG 200, a nontoxic and pharmacologically accepted solvent. In fact, the choice of A3B and A2B2 isomers for our studies was justified because they ensure a balance between good cellular localization, the generation of good singlet oxygen yields for efficient PDT, and good solubility in solvents accepted for pharmaceutical formulation.

Regarding the relationship between the structural and photophysical profiles of the A3B type porphyrins, studies show that structural asymmetry induces slight changes in the values of singlet oxygen formation quantum yield, fluorescence emission quantum yield, and the fluorescence lifetime of PS relative to symmetrical structures (Table 3) [162,172,179].

Furthermore, a comparative study on the photophysical behavior of 5,10,15,20-tetrakis(3-hydroxyphenyl) porphyrin and the corresponding reduced form 5,10,15,20-tetrakis(3-hydroxyphenyl) chlorin (active substance of Foscan^®^) highlighted very small differences between the photophysical parameter values (Φ_Δ_, Φ_F_, and τ_F_) (Table 3) [180]. The date presented in Table 3 confirmed that the photophysical properties of PS molecules can be modified by the coordination of metal ion at the porphyrinic ring, and changes related to electronic structure of the metal ion [159,160,162,172,179].

Regarding the potential for the internalization of A3B and A2B2 porphyrins at the cellular level, in vitro studies on various cell lines confirmed their localization in the cellular environment depending on the cell type, dose of the compound, and loading time [160,161,168]. As example, a laser scanning microscopy image of 5-(4-hydroxy-3-methoxyphenyl)-10,15,20–tris(4-acetoxy-3-methoxyphenyl)porphyrin (10 µM) uptake by human HT-29 colon carcinoma cells, is presented in Figure 5 [160].

In vitro biological studies performed on various cell lines (human colon carcinoma HT-29 cells, mouse L929 fibroblasts, peripheral blood mononuclear cells, human normal dermal HS27 cells, HaCaT keratinocytes, human peripheral blood SC monocytes) confirmed good biocompatibility for these asymmetrical structural configurations [160,161,168]. Furthermore, new amphiphilic porphyrins generated important amounts of singlet oxygen when activated with light at 635 nm [161,181].

In a recent study, one of the A3B-type asymmetric structures synthesized by us was investigated in vitro for its ability to reduce the number of tumor cells by PDT. In addition, the profile of stress gene expression changes triggered in vitro by porphyrin PDT in HT-29 human colon carcinoma cells was established [181].

By in vitro experimental PDT performed with TMAPMOHp, we observed the concomitant activation of particular cellular responses to oxidative stress, hypoxia, DNA damage, proteotoxic stress, and inflammation. This web of interconnected stressors underlies cell death, but can also trigger protective mechanisms that may delay tumor cell death or even defend cells against the deleterious effects of PDT. This stress-related molecular profile allowed us to identify potential therapeutic targets, such as the cytoprotective transcription factor NRF2, which could be used to develop co-therapies aimed at increasing PDT efficacy in cancer cells or protecting normal tissues [181].

### 4.2. Functionalizing of Porphyrin Type Macrocycles with Fragments of Bioactive Molecules

There are several strategies for improving the clinical performance of porphyrins, such as using a modified derivative design or specific delivery vehicles. The functionalizing of porphyrin macrocycles with moieties of carbohydrate, amino acids, antibody, and peptide, or their encapsulation into liposomes, micelles and nanoparticles, have been highlighted as a main strategy in development of third-generation PS.

The most interesting systems with photosensitizing properties nowadays are developed using natural resources, such as chlorophyll derivatives (chlorin compounds), which have a similar structure to hemoglobin and can be preferentially located in tumor cells [55,56,57]. The research group led by Li synthesized and evaluated a new photosensitizer, 5,10,15,20-tetrakis[(5-diethylamino)pentyl] porphyrin (TDPP) (Figure 6a), which has great potential in generating singlet oxygen and considerable antitumor potential with esophageal tumor cells (Eca-109) [154].

A new porphyrin derivative bearing ethylenediaminetetraacetic acid (Figure 6b), has demonstrated an intense phototoxicity effect on HepG2 and BGC823 cell lines and the destruction of tumor cells by lysosomal photodamage in vitro [156]. Fragments of isoquinoline attached to tetrapyrrole macrocycles (Figure 6c), have been shown to have effects on increased singlet oxygen efficiency and phototoxicity in relation to HT29 tumor cells [182].

Recent studies have pointed out the considerably increased generation efficiency of singlet oxygen correlated with great cell permeability and strong phototoxicity, as a result of attaching glucopyranoside moieties to the tetrapyrrole macrocycle and introducing In(III) ion inside the macrocycle (Figure 6d) [157].

Compared to the porphyrins, photosensitizers with chlorin structure have the advantage of intense absorption in the NIR spectral field, with considerable therapeutic relevance through the photosensitizing effect. The major disadvantage of some chlorin structures is weak solubility in water and biological fluids. That is why the second-generation chlorin photosensitizers are structurally modified by conjugating with a functional group of amino acids, peptides, or sugars.

The research group led by Smith developed a new photosensitizer derived from chlorin e-6 (e) by attaching lysine and aspartate fragments by a regioselective synthesis approach (Figure 7a) [183]. To obtain structures with good cell permeability and antitumor efficiency in K562 and P388 tumor cell lines, Guschchina et al. introduced hydrophobic amide groups into the chlorin e-6 structure (Figure 7b) [184].

Carbohydrates and their derivatives are attractive biomolecules for conjugating with PS, because they have the dual role of targeting and increasing water solubility. A water-soluble chlorin structure (Figure 8a) with excellent biocompatibility, absorption in the phototherapeutic field, and considerable phototoxicity was recently synthesized with four perfluorinatedphenyl rings and the attachment of maltotriose fragments [185].

The research group led by Pereira [186] reported the synthesis and evaluation of two isomeric porphyrin-galactose conjugates and their Zn(II) analogs (Figure 8b), with excellent potential in PDT against HCT-116 colon, MCF-7 breast, UM-UC-3 bladder and HeLa cervical cancer cells.

The introduction of new theranostic agents with good clinical efficiency is currently achieved by using advanced strategies to optimize the structural design and photophysical profile of porphyrin macrocycles, which includes attaching polyethylene glycol to them to obtain efficient nanostructures for the detection and treatment of tumor formations. Modern structural analyses of such systems have shown that they have a spherical architecture with a diameter of not more than 20 nm and an extinction coefficient around 7.8 × 10^7^ M^−1^ cm^−1^ [187].

### 4.3. Functionalizing of Porphyrin with Metal-Based Nanoparticles

An alternative to using structural modifications to improve the biodistribution of PS and increase its accumulation at tumor sites is to incorporate PS in the transport agents (liposomal systems, micelles, or nanoparticles [163,168].

Nanomedicine is an ever-growing field with huge potential for future discoveries and substantial progress in medicine, especially in cancer theranostics. In recent years, interdisciplinary studies have pointed out that translational nanotheranostics are a promising way forward in medicine. Biotech, pharmaceutical, and medical sciences companies have been active in its evolution, and are dynamic collaborators with researchers, governments, and educational institutions in developing and translating cancer nanomedicine. Two iron oxide contrast agents are now on the market, Ferridex and Gastro MARK from Advanced Magnetics, Inc., Cambridge, MA., and according to the analyses performed on several metal-based NPs (iron oxide, magnetic NPs, gold nanorods, colloidal gold) for imaging/theranostics, most clinical trials in nanomedicine are confined to polymer/lipid NPs, probably due to concerns regarding the adverse effects and/or toxicity of metal-based NPs [188].Therefore, there is a need to develop nanotheranostic tools and associated strategies, complemented by comprehensive toxicological studies, in order to find solutions for reducing the counteracting of clearly defined toxicity issues, while exploiting their huge potential as theranostic agents.

Polymer-coated NPs can act as carriers for porphyrins. Metal-based NPs carry an additional theranostic property: irradiation with near infrared (NIR) light will activate the covering photosensitizer in PDT and trigger a concurrent photothermal effect, mediated by the associated metal. This will lead to the enhanced power of the nanosystems for tumor cell ablation and/or a vasodilatation-mediated increase in NP accumulation in tumors. Moreover, the same NIR radiation used to activate porphyrinic photosensitizers can induce additional ROS production by direct interaction with the metal core of NPs. By their excellent photophysical and photochemical properties, porphyrins can be good generators of tumor-destructive ROS in PDT. Additionally, the fluorescence of the compounds allows the monitoring of NS uptake by cancer cells, thus improving their imaging power (combined magnetic and fluorescence characteristics). As porphyrinic compounds preferentially accumulate in tumors, a concentration of NS in the diseased tissue can be achieved, the therapeutic dose can consequently be lowered, and unwanted side effects could be partially avoided.

To overcome the drawbacks related to the aggregation and low solubility of PS molecules in biological media, in recent studies, functionalized metal-based nanocarriers were used to deliver porphyrin to tumors [189,190,191,192]. Moreover, coating inorganic NPs with biocompatible polymers (PEG, chitosan, alginate, heparin, cellulose, gelatin, etc.) can reduce particle uptake by phagocytes, promoting the stabilization and biocompatibility of the system and increasing the blood circulation time. Hydrophilic polymeric chains can be physically linked to porphyrins in order to enhance porphyrin attachment and the biocompatibility of NP/NS. For metal NP synthesis, several methodologies can be used, including co-precipitation. Porphyrins can be attached to NPs by means of appropriate linkers. Applying a thin silica layer on the surface of NPs greatly improves the conjugation of porphyrin on the functionalized NP surface.

Recently, Vieira Ferreira and collab. [189,190] developed superparamagnetic iron NPs coated with a porphyrin derivative. The surface photochemistry of two tetrapyrrolic structures was reported as TCMP (Figure 9) and TCMPMOH_P_, covalently bound to silica-coated magnetite nanoparticles (Fe_3_O_4_@SiO_2_) and imbibed into a polyethylene glycol (PEG) matrix, in the form of a fine powder.

Two populations of porphyrin molecules exist, free porphyrin (unquenched) and porphyrin with Fe_3_O_4_@SiO_2_ NPs attached (quenched). In this way, the amount of available excited porphyrin for singlet oxygen formation is reduced, thus the presence of Fe_3_O_4_@SiO_2_ NPs may not favor this specific PDT action by the type II mechanism, but it prevents aggregation of the porphyrins, even when a PEG matrix is used, and type I mechanisms remain active. In conclusion, the possibility of loading porphyrinic molecules covalently bound to ferromagnetic nanoparticles represents a great advantage in nanotechnology, because it avoids the major problems related to the aggregation and low solubility of PS in biological media; moreover, this approach allows the establishment of an accurate diagnosis and the optimization of the therapeutic outcome.

## 5. Conclusions

Tetrapyrrolic compounds are among the most studied with regard to their antitumor potential, because their versatile structure and spectral profile make them applicable as photosensitizers.

At present, one of the primary objectives of research in the pharmaceutical and oncological fields is to establish an optimum protocol for the identification and treatment of tumor formations by using molecules with exceptional pharmacological properties and minimal side effects.

In this review, we summarized the theoretical aspects regarding the structural, spectral, and biomedical profiles of porphyrin macrocycles. The bibliographical data concerning the properties and theranostic potential of this tetrapyrrole type are complemented by some ideas related to strategies for improving the biodistribution of PS at tumor sites. Our research group synthesized and characterized a panel of new amphiphilic porphyrins exhibiting structural and spectral profiles suitable for theranostic applications. “Green chemistry” (solvent-free reactions activated by microwave irradiation) was applied for the synthesis of amphiphilic porphyrins. We can appreciate the fact that extensive interdisciplinary research, accompanied by detailed toxicological assessments, can guide researchers in finding the most appropriate variants for the development of new tetrapyrroles as theranostic agents. Furthermore, the mechanisms of action at the cellular level should be thoroughly investigated for scientific documentation of the balance between benefits and side effects. Finally, we anticipate that the information presented in this review will encourage the design and development of new porphyrin macrocycles as theranostic agents.

## Figures and Tables

**Figure 1 molecules-28-01149-f001:**
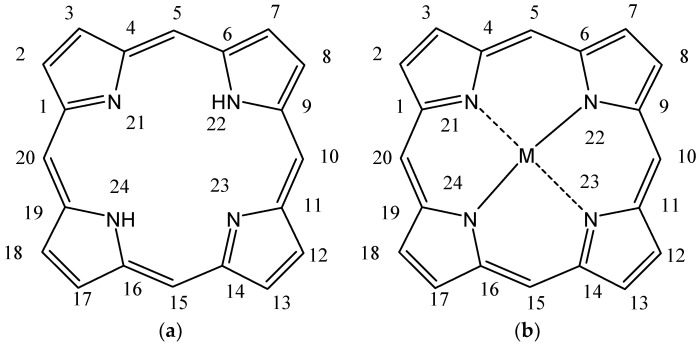
General structures of (**a**) free base porphyrin and (**b**) metalloporphyrin (atoms numbered according to IUPAC (https://doi.org/10.1351/goldbook). (accessed date: 17 January 2023).

**Figure 3 molecules-28-01149-f003:**
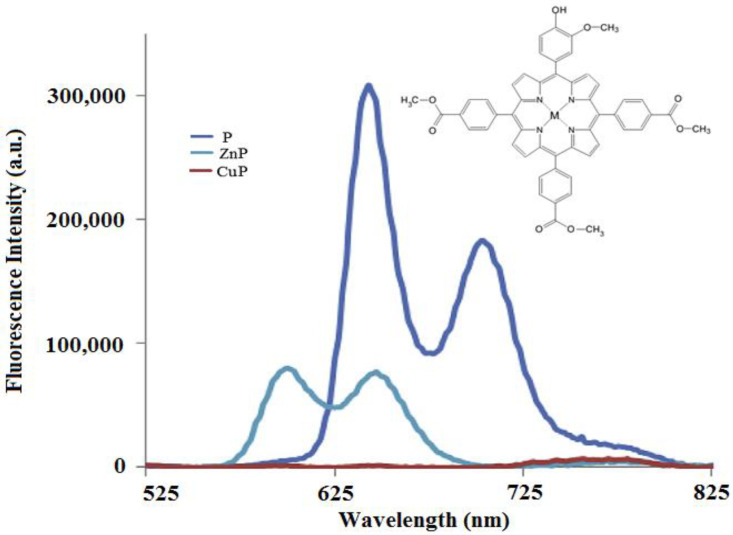
Fluorescence emission spectra for 5-(4-hydroxy-3-methoxyphenyl)-10,15,20-tris(4-carboxymethylphenyl) porphyrin (P), 5-(4-hydroxy-3-methoxyphenyl)-10,15,20-tris(4-carboxymethylphenyl)porphyrinatozinc(II) (ZnP) and 5-(4-hydroxy-3-methoxyphenyl)-10,15,20-tris(4-carboxymethylphenyl)porphyrinatocopper(II) (CuP) (ethanol used as solvent) [23].

**Figure 4 molecules-28-01149-f004:**
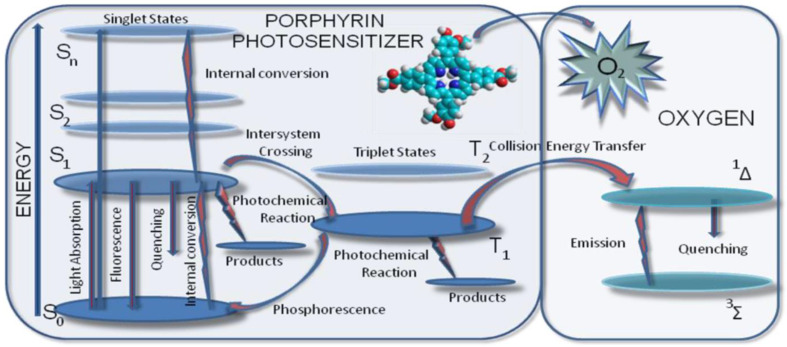
Jablonsky diagram (S_0−_ground state of photosensitizer; S_1_, S_2_, excited singlet states of photosensitizer; T_1_, excited triplet state of photosensitizer).

**Figure 5 molecules-28-01149-f005:**
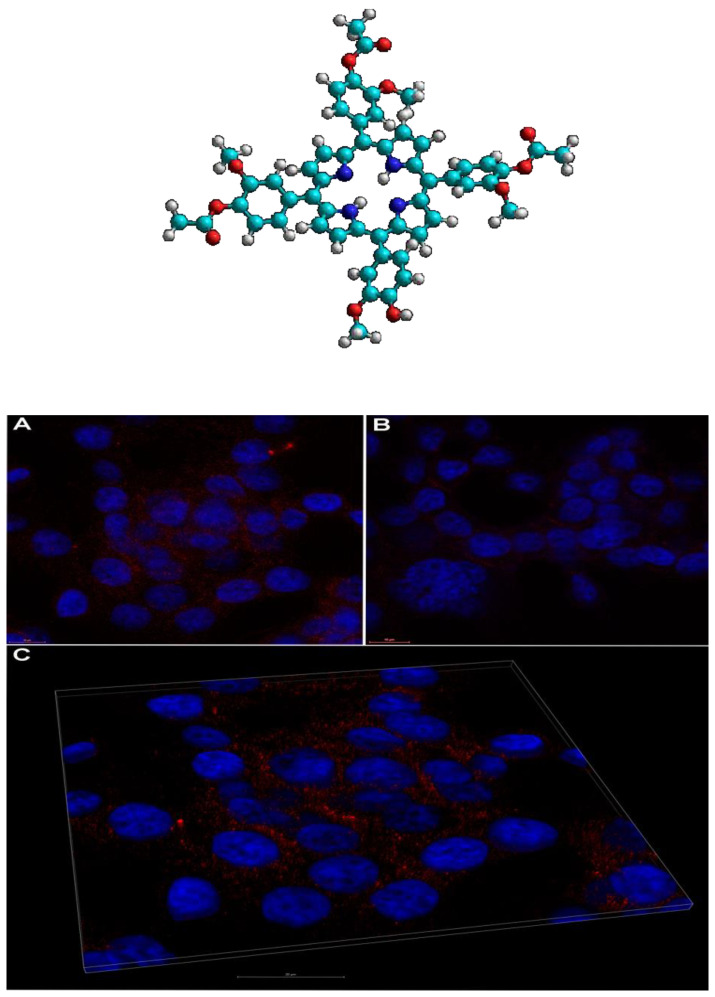
Representative laser scanning microscopy image of 5-(4-hydroxy-3-methoxyphenyl)-10,15,20–tris(4-acetoxy-3-methoxyphenyl) porphyrin (10 µM) uptake by human HT-29 colon carcinoma cells: (**A**): porphyrin (red) scattered throughout cytosol compared to control (**B**); (**C**): 3D volume rendering of 2.36 μm z-stack from A; nuclei were stained with DAPI (blue); scale bar 10 μm in (**A**,**B**) and 20 μm in (**C**) [160].

**Figure 6 molecules-28-01149-f006:**
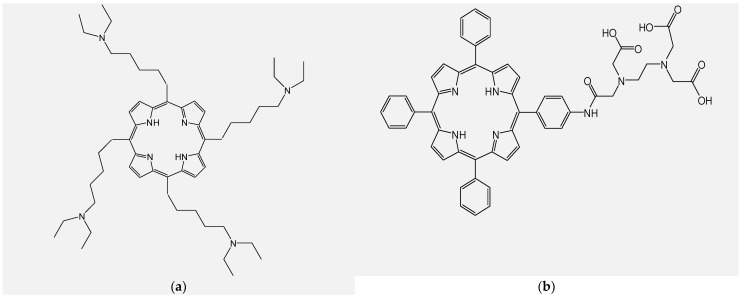
Structures of porphyrin type macrocycles functionalized with fragments of diethylaminopentyl alchol (**a**) [154], ethylenediaminetetraacetic acid (**b**) [156], isoquinoline (**c**) [182], glucopyranoside (**d**) [157].

**Figure 7 molecules-28-01149-f007:**
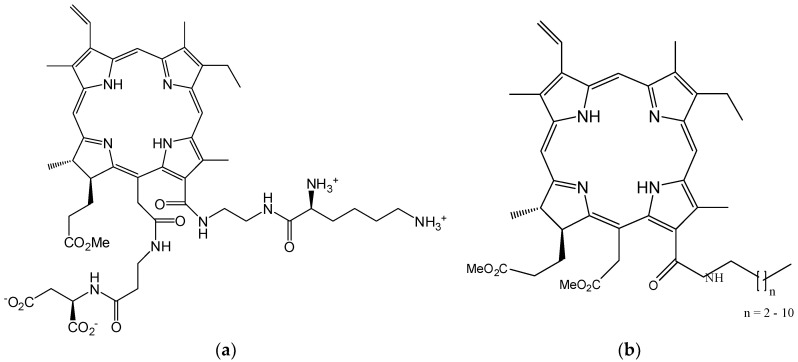
Structures of chlorin type macrocycles functionalized with lysine and aspartate fragments (**a**) [183] or hydrophobic amide groups (**b**) [184].

**Figure 8 molecules-28-01149-f008:**
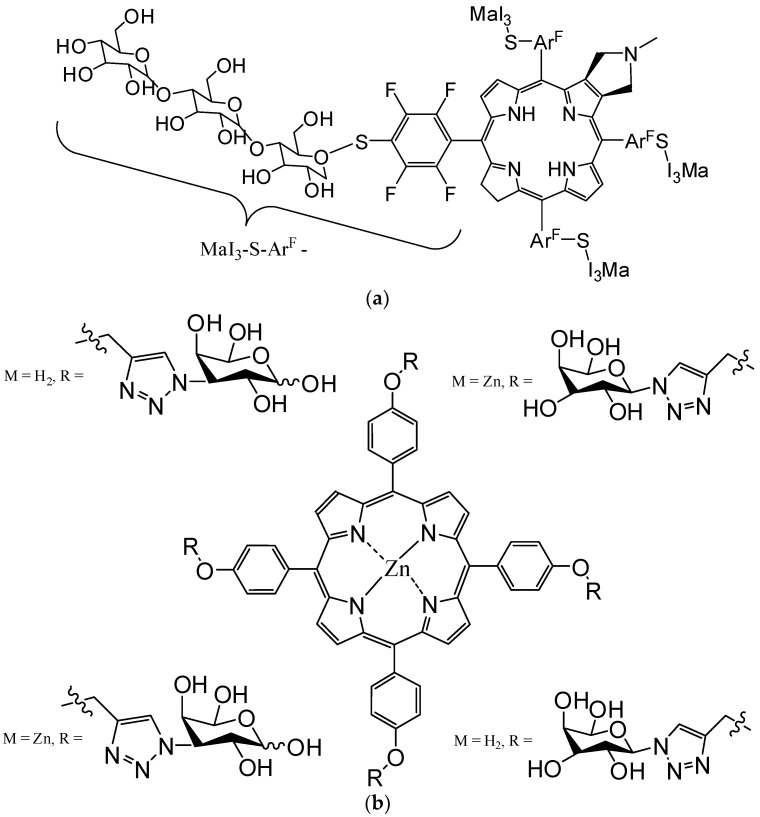
Structures of macrocycles photosensitizers functionalized with maltotriose (**a**) [185] or galactose (**b**) [185] fragments.

**Figure 9 molecules-28-01149-f009:**
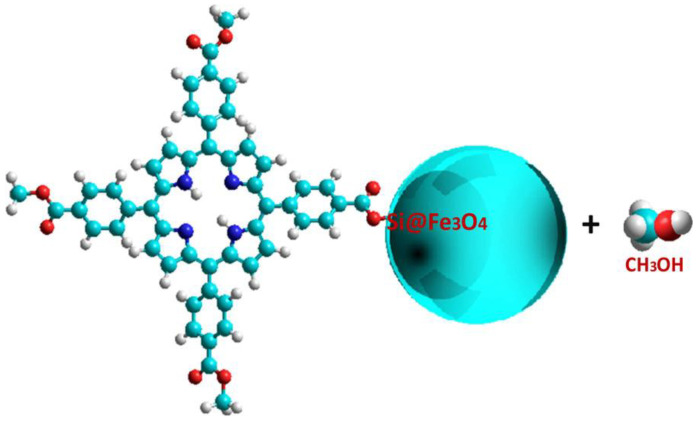
New covalent bond formed after transesterification reaction of TCMP with silanol groups of the surface of silica-coated magnetite nanoparticles [190].

**Table 1 molecules-28-01149-t001:** General presentation of photosensitizers with macrocycle structures approved or in clinical trials for diagnosis and photodynamic therapy of cancer [99,100,101,102,103,104].

Photosensitizer	Active Substance, Activation Wavelength *λ* (nm)	Clinical Approval
PHOTOFRIN^®^ (Axcan Pharma Inc., Mont-Saint-Hilaire, QC, Canada)	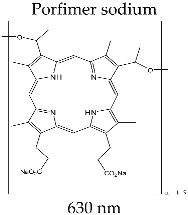	Theranostic for tumor formations in lung, brain, cervix; therapy for bladder and esophagus cancer; in clinical trial for brain cancer diagnosis
FOSCAN^®^ (Biolitec Pharma Ltd., Jena, Germany)	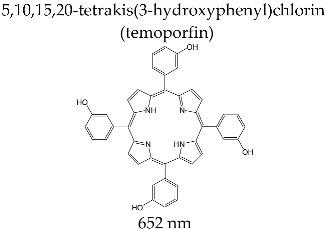	Photodynamic therapy for carcinoma with squamous cells at late stage in head and throat level; diagnosis of brain, bladder, and ovarian tumors; in clinical studies for therapy for breast, pancreas, prostate, lung, stomach, and skin cancer
LASERPHYRIN^®^ (Meiji Seika Pharma Co., Ltd., Tokyo, Japan)	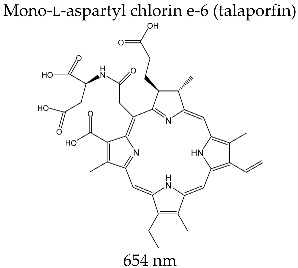	Photodynamic therapy for lung and brain cancer; intraoperative photodiagnosis of malignant brain formations; clinical studies for treatment of colorectal and liver neoplasms
VISUDYNE^®^ (Novartis Pharmaceuticals, East Hanover, NJ, USA)	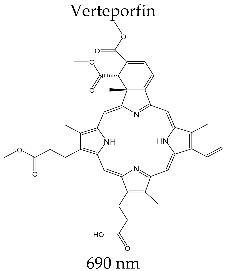	Photodynamic therapy for macular exudative degeneration with subfoveal choroidal neovascularisation of melanoma and psoriasis; clinical trial for diagnosis of ovarian tumor formation
PHOTOCHLOR^®^ (Roswell Park Cancer Institute, Buffalo, NY, USA)	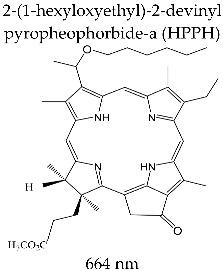	Clinical trials for therapy (first and second stage) of cervical intraepithelial neoplasia, esophagus tumor, skin, lung, and oral cavity cancer; clinical trials of marker potential in identifying different types of cancer
PURLYTIN^®^ (Miravant Medica Technologies, Santa Barbara, CA, USA)	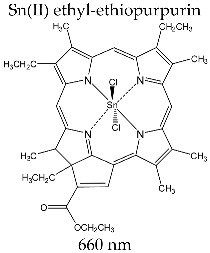	Clinical trials for photodynamic therapy of breast adenocarcinoma, Kaposi’s sarcoma, prostate cancer, cerebral metastasis, and psoriasis
TOOKAD^®^ (WST09) (Negma Lerads/Steba Biotech, Toussous le Noble, France)	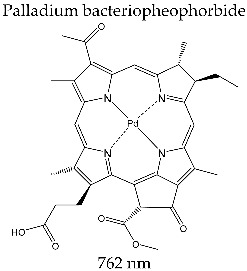	Stage II/III clinical trials targeting photodynamic therapy for prostate cancer
LUTRIN^®^ (Pharmacyclics Inc., Sunnyvale, CA, USA)	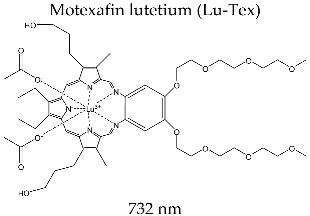	Clinical trials for prostate, breast, cervical, and brain cancer, melanoma, and Kaposi’s sarcoma
PHOTOSENS^®^ (SSC NIOPIK, Moscow, Russia)	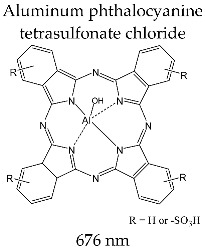	Clinical trials for gastric, oral, skin and breast cancer

**Table 2 molecules-28-01149-t002:** Unsymmetrical A3B and A2B2 type *meso*-substituted porphyrins.

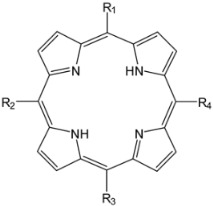 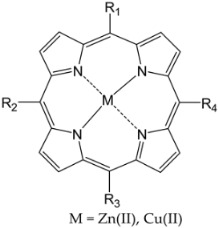
A3B type*meso*-substituted porphyrins	R_1_	R_2_ = R_3_ = R_4_	Ref.
TCMPMOHpM(II)TCMPMOHp	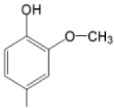	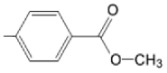	[31]
TMAPMOHpM(II)TMAPMOHp	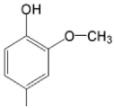	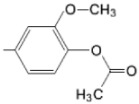	[160]
TMAPDOH	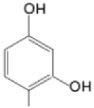	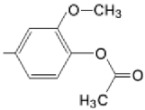	[161]
TCMPOHoM(II)TCMPOHo	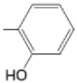	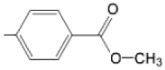	[162]
TRMOPPM(II)TRMOPP	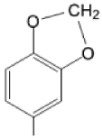	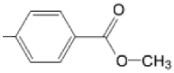	[163,172]
TCMPOHpM(II)TCMPOHp	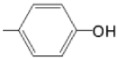	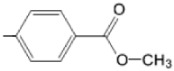	[164,170]
TCMPOMoM(II)TCMPOMo	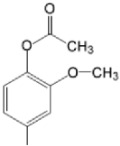	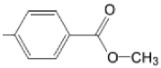	[165,172]
TCMPOHmM(II)TCMPOHm	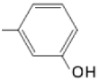	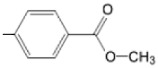	[166,175]
TMAPOHmM(II)TMAPOHm	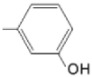	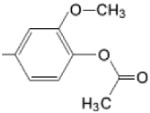	[167,178]
TMAPOHoM(II)TMAPOHo	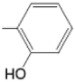	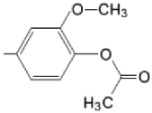	[167,178]
M(II)TMAPOHp	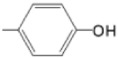	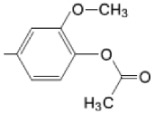	[171]
A2B2 type *meso-*substituted porphyrins	R_1_ = R_3_	R_2_ = R_4_	
TCMPDMOH	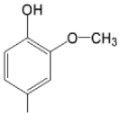	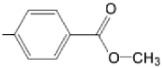	[168]

**Table 3 molecules-28-01149-t003:** Singlet oxygen formation quantum yield (Φ_Δ_), fluorescence emission quantum yield (Φ_F_), and fluorescence lifetime (τ_F_) for a series of tetrapyrrolic compounds (solvent CHCl_3_, except *marked structures were CH_3_OH was used).

Tetrapyrrolic Compound		Ref.
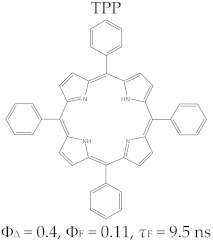	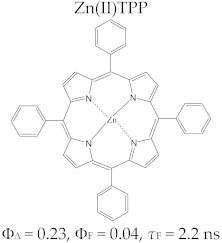	[162]
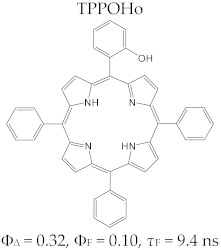	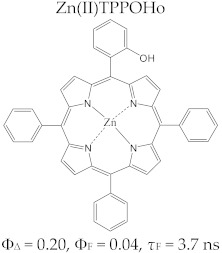	[179]
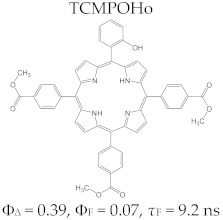	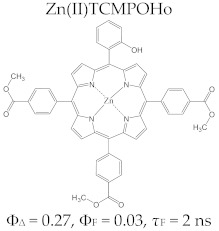	[162]
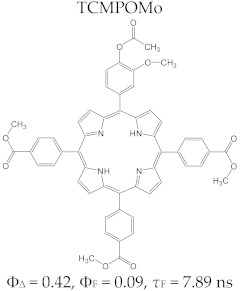	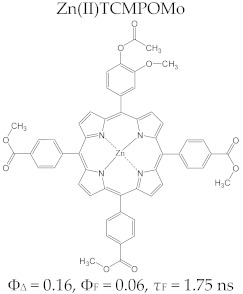	[172]
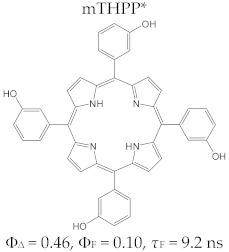	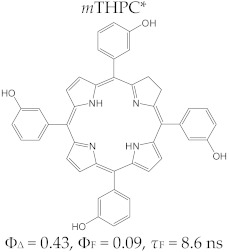	[180]

## Data Availability

Not applicable.

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
