# Peer review of "Porphyrin Macrocycles: General Properties and Theranostic Potential"

_molecules, 2023, doi:10.3390/molecules28031149_

Round 1

Reviewer 1 Report

molecules-2139101

In this manuscript, the authors review properties and potential phototherapeutic applications of porphyrin and its analogues. The authors summarize currently approved porphyrinic photosensitizers for diagnosis and PDT therapy for cancers.

General comments:

(1)   The authors should make all molecular structures throughout the manuscript uniform.

(2)   Use abbreviations for all the listed compounds for porphyrins throughout the paper as it is not necessary to use the full names for reading purposes. Instead, the authors can add something like an appendix that list the full names for each compound mentioned in the review.

(3)   Use super- and subscript properly.

(4)   The authors are encouraged to check the use of certain scientific words throughout the manuscript. For example, people generally use ground state that refers to S0 and excited state that refers to Sn or Tn. Please check the whole manuscript carefully.

(5)   The authors need to elaborate more on what distinguish porphyrin than other photosensitizers in the PDT for cancers and highlight more on what are the futures for porphyrins in this field.

Specific comments:

(1)   Cite more broadly on the comments made from line 43 to 44.

(2)   Cite more broadly on the porphyrin properties from line 51 to 54. The authors should cite more review articles and books on porphyrins especially The Handbook of Porphyrins.

(3)   What are these I, II, III, and IV referring to on line 112?

(4)   References needed on line 120.

(5)   The paragraph from line 145 to 149 seems redundant as it was discussed already above.

(6)   On line 157, I would not use fluorescent signal, instead, I will use quantum yields.

(7)   The statement made from line 162 to 163 about Pd porphyrin can’t be true as Pd largely quench porphyrin centered singlet excited states, so it can’t increase the lifetime and quantum yield for fluorescence.

(8)   Line 208, see general comments. Use ground state and excited state.

(9)   What does it mean “Stabilization of the S1 state can be reached by the emission of photons” from line 210? I don’t understand.

(10) What does it mean “their administration is difficult and accompanied by a molecular aggregation phenomenon”

(11) Line 294, the author should make it clear that not only chlorins suffer from aggregation in solutions but all the tetrapyrrole macrocycles have the same problem.

(12) The authors should cite more broadly on water-soluble chlorins from Professor Jonathan S. Lindsey.

Conclusion:

This manuscript needs a major revision and has to be reviewed again before the decision can be made.

Reviewer 2 Report

Authors in review entitled:" Porphyrin Macrocycles: General Properties and Theranostic Potential" focus on studies of porphyrin macrocycles in terms of their structural and spectral profiles relevant to their applicability in efficient cancer diagnosis and therapy. The topic and the application of selected molecules and formulations are interesting. However, the manuscript should be substantially revised

Main comments:

Figures 2 and 3 are not mentioned in the main text.

Section 3 should describe theoretical aspects on porphyrins as photosensitizers. However, this section describes historical aspects of light application, photodynamic processes, and cell death (apoptosis and necrosis). Nothing is said here about porphyrins. This section should be corrected to be more specific to porphyrins. This general aspect of light and PDT is well known and described in many reviews.

Page 16, lines 615-624 is confusing. The authors wrote a review paper on general aspects and suddenly mentioned here some A3B and A2B2 compounds, which the reader will recognize only later from a table/figure not presented in the text. So, is this an original study or a review paper?

The graph in panel (a) of Figure 5 shows the mean values of the intensities but lacks the error bars.

The images in panel (b) of Figure 5 are not well presented. This is not acceptable for publication. In addition, the description in the legends of A-B is not clearly seen from images.

Section 5. Strategies aimed at improving the therapeutic potential of porphyrin photosensitizers presents special cases and a summary of products developed by the authors and does not provide a strategy as mentioned in the subtitle.

In general, this review is inconsistent and should be better elaborated. There are many separate topics that are not connected. The topic chosen by the authors is interesting, but the consistency of the manuscript is not homogeneous. The authors should focus more on the topics mentioned in the subtitles, and some can be more divided.

Round 2

Reviewer 1 Report

The current version is suitable for publication in Molecules.

Author Response

Thank you for your prompt acceptance of our 1sr Revised Manuscript

Reviewer 2 Report

The manuscript was significantly improved.

Author Response

Thank you for your prompt comment on our 1st Revised Manuscript